# Spatial and Temporal Analysis of COVID-19 Cases in West Java, Indonesia and Its Influencing Factors

**DOI:** 10.3390/ijerph20043198

**Published:** 2023-02-11

**Authors:** Delima Istio Prawiradhani Putri, Dwi Agustian, Lika Apriani, Ridwan Ilyas

**Affiliations:** 1Epidemiology Study Program, Faculty of Medicine, Universitas Padjadjaran, Jalan Eyckman No. 38 Gedung RSP Unpad Lantai 4, Bandung 40161, Indonesia; 2Division Epidemiology and Biostatistics, Department of Public Health, Faculty of Medicine, Universitas Padjadjaran, Jalan Ir. Soekarno KM. 21, Jatinangor, Sumedang 45363, Indonesia; 3Informatics Department, Faculty of Science and Informatics, Universitas Jenderal Achmad Yani, Jalan Terusan Jenderal Sudirman, Cimahi 40531, Indonesia

**Keywords:** COVID-19, Indonesia, spatial pattern, temporal, vaccine COVID-19

## Abstract

Coronavirus Disease 2019 (COVID-19) spread quickly and reached epidemic levels worldwide. West Java is Indonesia’s most populous province and has a high susceptibility to the transmission of the disease, resulting in a significant number of COVID-19 cases. Therefore, this research aimed to determine the influencing factors as well as the spatial and temporal distribution of COVID-19 in West Java. Data on COVID-19 cases in West Java obtained from PIKOBAR were used. Spatial distribution was described using a choropleth, while the influencing factors were evaluated with regression analysis. To determine whether COVID-19s policies and events affected its temporal distribution, the cases detected were graphed daily or biweekly with information on those two variables. Furthermore, the cumulative incidence was described in the linear regression analysis model as being significantly influenced by vaccinations and greatly elevated by population density. The biweekly chart had a random pattern with sharp decreases or spikes in cumulative incidence changes. Spatial and temporal analysis helps greatly in understanding distribution patterns and their influencing factors, specifically at the beginning of the pandemic. Plans and strategies for control and assessment programs may be supported by this study material.

## 1. Introduction

In December 2019, severe and fatal cases of the new SARS-CoV-2 virus were reported in Wuhan, China [1]. Similar cases quickly emerged in other countries, leading the WHO to declare a COVID-19 pandemic on 11 March 2020 [2]. Indonesia, with the fourth largest population in the world, was heavily affected by the pandemic. Additionally, on 15 July 2021, Indonesia reported the highest daily number of new positive COVID-19 cases globally, mainly in DKI Jakarta and West Java, which is home to about one-fifth of Indonesia’s population and serves as an economic hub for the country [3,4]. 

The Indonesian government initially denied the existence of COVID-19 in the country in early 2020 but declared a state of national emergency in late January 2020. To contain the spread, mitigation policies were enacted, such as restricting public activities and isolating new cases at home or in special facilities and hospitals. Despite the pandemic, the government did not impose a complete lockdown due to economic reasons; instead, various levels of public restrictions were implemented based on local transmission. Amidst the limitations in the healthcare system, geographical challenges, and disparities in infrastructure and healthcare worker availability, the government attempted the quick implementation of a mass vaccination program to raise herd immunity, by mobilizing the army and police. Despite these efforts being based on the growing literature regarding COVID-19, their effectiveness in reducing morbidity and mortality is uncertain. 

Social distancing, quarantine and case isolation, international border closure, public activity lockdown, and mask-wearing are among the non-pharmaceutical measures that reduce SARS-CoV-2 transmission in the absence of pharmaceutical measures such as anti-virus drugs and vaccines. Therefore, concerning the spread of the re-emerging virus, the temporal and spatial dynamics of COVID-19 largely depend on how promptly and effectively local authorities can implement control measures [5,6,7]. Studies from different countries have shown that a low compliance with social distancing and public activity restrictions leads to a rapid increase in morbidity and mortality, while strict lockdowns and high compliance reduce transmission as well as the number of cases and fatalities [8,9,10]. The availability of vaccines provides additional protection by accelerating the attainment of herd immunity through mass vaccination programs for susceptible populations. The Indonesian government has implemented such programs, but there are no studies to analyze their effect on the spread of COVID-19 [11]. 

Based on the descriptions above, this research aims to analyze the history of COVID-19 spread and determine the factors driving its distribution by integrating and interpreting existing quantitative and qualitative data. This approach is expected to provide a deeper insight into the dynamics of pandemics as well as facilitate well-informed decisions and the formulation of evidence-based health policies in mitigating similar future pandemics. Additionally, this research seeks to demonstrate the value of spatial analysis in policy-making, particularly for the control and prevention of infectious diseases [10,12,13,14,15,16,17]. 

## 2. Materials and Methods

This study used ecologically mixed methods with a convergent approach that combined both quantitative and qualitative methods to answer the research question [18]. The quantitative method examined the factors affecting the spatial distribution of cases, while the qualitative was used for the temporal distribution. The study population was the population of West Java Province, where the spatial analysis unit was a geo-political area represented as a district/municipality in West Java Province, Indonesia. West Java Province is located side by side with the Jakarta City, the capital of Indonesia. The daily/bi-weekly number of new cases for the whole province was employed in the temporal analysis. 

### 2.1. Materials

The PIKOBAR (West Java Province Government COVID-19 Information and Coordination Center) database was the main source of the quantitative data collected on the number of cases and vaccinations. The population size and area information were obtained from the central statistics agency’s website for West Java (https://jabar.bps.go.id/; accesed 21 February 2022). The data on government regulations and policies for the qualitative research of temporal distribution were obtained from the West Java provincial website (jabarprov.go.id; accesed 21 March 2022) and Indonesia’s COVID-19 task force (https://covid19.go.id; accesed 22 March 2022).

### 2.2. Methods

#### 2.2.1. Data Cleaning and Transformation

Row de-identified data without personal identities were extracted from the PIKOBAR database for a case list that includes dates and addresses at the district/municipality level. It was aggregated by time (bi-weekly and daily) and space (district/municipality) for temporal and spatial analysis, respectively. The vaccination data were also extracted from a different table within the PIKOBAR database and aggregated by date and district/municipality. The population size and area of each district/municipality were also obtained from the database to form a separate table. All data were integrated into a master spatial table using official district/municipality identification numbers in QGIS software version 3.18.0—Zürich. The policy and event data were collected from the West Java and Indonesia’s COVID-19 websites, tabulated, and categorized for temporal analysis. To improve the clarity of the processes, all data cleaning and transformation steps are detailed in Appendix A. 

#### 2.2.2. Variables

The outcome variables for this research were the number of cases and incidence. The number of cases was used for temporal analysis and focuses on the transmission dynamics over time in the same location. Meanwhile, the incidence was used for the spatial analysis by considering the number of people at risk in each district and is calculated as the number of cases in a certain period and 30 days after vaccination, divided by the population in the area, standardized to 100 and 100,000 persons, respectively. The explanatory variables consisted of population density and vaccination coverage. The population density is the number of people per square kilometer, and vaccination coverage is the percentage of people who have received at least one dose of the COVID-19 vaccine in each district. 

#### 2.2.3. Data Analysis 

(1)Temporal Analysis

Biweekly case charts were created to illustrate the overall temporal dynamics of COVID-19 in West Java Province. For a further investigation of the possible influencing factors, daily case charts were also produced with historical events, policy implementation, and new virus variants.

(2)Spatial Analysis

First, an exploratory spatial data analysis (ESDA) was performed to reveal the spatial pattern and heterogeneity of COVID-19 incidence as the outcome variables, while population density and vaccination coverage were the explanatory variables or covariates. At this step, choropleth maps were created for the mentioned variables to show the spatial distribution of both disease and exposures. Moran’s I and local indicator of spatial autocorrelation (LISA) statistics were employed to detect global and local clustering patterns of the disease, respectively. Based on the ESDA result, regression models were selected with an algorithm developed by Luc Anselin [19] to further investigate possible factors associated with COVID-19 incidence. To evaluate the impact of vaccination, two alternative cumulative incidences (one week and two weeks) as the outcome variables were calculated after 30 days of vaccination cumulative coverage. The statistical results with a *p*-value < 0.05 were considered significant. All maps were created using the QGIS software, while Moran’s I, LISA, and regression analysis were conducted using GeoDa application version 1.18.0. 

## 3. Results

### 3.1. Temporal Distribution of New COVID-19 Cases

The biweekly temporal dynamics of the virus transmission are demonstrated in Figure 1 to depict a general overview of the research period. This showed two pandemic waves, where the second (July 2021) was above three folds higher than the first (January 2021). The 369,765 new cases detected during the second wave span, which started in early June 2021 and lasted until August 2021, contributed to 52.4% of the total cases recorded.

Two COVID-19 temporal charts with daily cases were created for a detailed analysis of the impact of events, holidays, and public health policies on the disease dynamics as presented in Figure 2 and Figure 3. 

### 3.2. Spatial Heterogeneity of COVID-19 Incidence

During the research, Kota Depok, Kota Cirebon, Kota Bogor, and Kota Bekasi had the highest COVID-19 incidence rates, ranging from 3.07 to 4.23 cases per 100 people. All areas classified as Kota (City) had a higher COVID-19 risk than average (1.62 per 100 people). Tasikmalaya had the lowest incidence rate at 0.4 cases per 100 people. Statistic of spatial variable as show in Table 1 and mapping in Figure 4. 

To further investigate the factors associated with COVID-19 incidence in each area, population density and vaccination coverage spatial distribution are presented in Figure 5 and Figure 6. These maps visually exhibit the spatial heterogeneity of both variables and may explain the variability in COVID-19 incidence across the areas.

The vaccination coverage is showed in Figure 6.

An additional map for the spatial distribution of COVID-19 incidence after the vaccination coverage calculation period is presented in Figure 7 and Figure 8.

Moran’s I statistics calculation for detecting global clustering of the disease produced a result of 0.051 with a *p*-value of 0.254, as presented in Figure 9 histogram of random permutation indicating that the general spatial distribution of COVID-19 incidence was random. 

Figure 10 shows that there were two significant areas of local clustering, depicted in dark blue for low–low areas and pink for high–low areas There is no High–High area which indicate a hot spot cluster and Low–High area, as indicated by the local spatial autocorrelation analysis. To describe variables in low–low core clusters we make a city table statistic in Table 2. 

### 3.3. Ecological Factors Associated with Spatial Distribution of COVID-19

The results of the multivariable linear regression analysis to determine the association between the covariates (population density and vaccination coverage does affect the spread of disease) and COVID-19 incidence with three model candidates are presented in Table 3.

## 4. Discussion

The COVID-19 pandemic wave in West Java during the research period had two prominent peaks in 2021, consistent with the overall trend in Indonesia [20]. This is probably because 20% of Indonesia’s population lives in West Java, which is located near Jakarta and the Cengkareng international airport, as shown in Figure 7. Therefore, the COVID-19 incidence in West Java is representative of the overall trend in Indonesia. 

In 2020, COVID-19 incidence was generally lower compared to 2021. Due to the nature of SARS-CoV-2 transmission that occurred through person-to-person physical interaction, the epidemic curve of the disease will be driven strongly by how effective the population mobility or travel restriction policy is implemented, particularly when vaccines are unavailable. This situation was relevant during the early phase of the pandemic in 2020, as public health authorities in West Java Province implemented restrictions on public activities due to the perceived high fatality rate of COVID-19. This may have reflected high community compliance and fear of the disease. However, this must be cautiously inferred as a limited testing and reporting capacity for COVID-19 in Indonesia may have contributed to the lower incidence [21,22,23]. Advanced countries such as the USA had more RNA molecular testing facilities, while countries with full lockdowns such as China and New Zealand [24,25,26,27] showed very low COVID-19 incidence and fatalities [20]. 

The Indonesian government never implemented a full lockdown policy due to economic reasons; instead, they relied on the COVID-19 vaccination programs which commenced in January 2021 to increase the herd immunity level for disease prevention [28,29,30]. The temporal analysis showed that the second wave of COVID-19 occurred in mid-2021 despite efforts to prevent it. This was due to the emergence of the delta variant in early May 2021, which was more virulent and easily transmitted [31,32]. Countries including China and New Zealand kept their COVID-19 incidence low or near zero by extending full lockdown policies [20]. Social distancing and travel restrictions were found to be effective in preventing the spread of COVID-19 based on mathematical models [33]. However, these policies were not effectively implemented in West Java and failed to prevent the second wave of high incidence in June 2021. This may have been due to low community compliance or a delay in policy implementation. 

The spatial analysis showed that COVID-19 incidence was randomly distributed, as indicated by the negative result of the global spatial autocorrelation test. This could be due to the modifiable areal unit problem (MAUP) in the spatial epidemiology field, where disease cases and populations were aggregated into the district boundary (Kabupaten/Kota) as the unit of analysis. This boundary choice may obscure the original epidemic pattern, for example, supposing a high incidence in one sub-district is offset by a lower incidence in others. To reduce the MAUP effect, a smaller areal unit (e.g., sub-district) for data aggregation is ideal, but this information was not available for this research. However, local clustering was detected for the low–low cluster (*p*-value of 0.01), covering two areas (Garut and Sumedang). This pattern raises questions about the cause, which could be ecological factors or merely data artifacts/noise, and all will be discussed further. 

To investigate the factors driving the spatial heterogeneity of COVID-19 incidence in West Java, three linear multivariable regression models were developed, which included two variables, namely, population density and vaccination coverage—dose 1. The second and third models, which limited the outcome to the period after vaccination, showed that both the population density and vaccination coverage level were significantly associated with the incidence at the district population level but in different directions. These findings support large-scale vaccine administration research, which discovered the ability of vaccines to reduce COVID-19 infection rates [34,35,36]. Based on the results, some time is required for the vaccine to have an impact on reducing the incidence, even after adjusting for population density. For every 1% increase in vaccination coverage, it is estimated that an additional five to six persons per 100 population are protected on average. Meanwhile, a decrease of 100 persons per kilometer square in population density decreases incidence by two persons per 100 of the population. The low–low core cluster areas (Garut and Sumedang) identified by the LISA were driven more by a low population density (first quartile rank) than vthe accination coverage level (second quartile rank). Population density is positively correlated with cumulative incidence, which is consistent with other studies that have found that COVID-19 spreads more easily through movement and human contact in urban areas with a higher population density [37].

The results obtained have several implications that ought to be noted. Timely travel restrictions and full lockdowns are required to prevent the spread and outbreak of viruses with high transmission rates and case fatality ratios, such as the SARS-CoV-2 delta variant [38,39]. This is consistent with other research that suggests that a high compliance with social distancing is needed to suppress the transmission of the delta variant [40] Contact tracing and isolation are less effective in stopping the fast transmission of the delta variant, which has a shorter infection generation time compared to the alpha variant [38]. Given the high fatality rate of the delta variant (estimated at 3.4%) [38,39], this would result in approximately 12,000 deaths in West Java alone over a 3-month period, which could be reduced by a timely lockdown policy. Therefore, we believe that lockdown readiness should be urgently included as a contingency plan in future pandemic preparedness.

In addition to the lockdown policy, vaccination could provide some protection by increasing the herd immunity level. Considering the delayed implementation in West Java, which might be due to logistic and supply constraints, vaccination programs were not sufficient to prevent the second pandemic wave caused by the emergence of the delta variant in this community. Therefore, a comprehensive mass-vaccination rollout program should preferably be conducted earlier in a pre-outbreak phase. This tends to play an important role in the longer term. Based on the evidence that population density is also associated with a higher incidence, under resource scarcity situations, the area with a higher population density should be more highly prioritized for COVID-19 vaccination programs. 

## 5. Strengths and Limitations

During the research period, there were some dynamic testing and reporting capacity issues, with the tendency to cause differential underreporting and create some bias in the disease incidence estimation over time and space or areas. Due to the lack of data to measure underreporting phenomena, this research relied on the assumption that the reporting bias was distributed randomly over time and space. It was also suggested that the travel restriction policy in West Java was ineffective in promoting social distancing and failed to prevent a surge in COVID-19 cases caused by the delta variant. This conclusion should be confirmed with the unincluded population mobility data. In future research, adding population mobility data from sources such as Waze or Google would be recommended. A limited sample size of 27 data points representing district or municipality areas was used, and COVID-19 incidence was calculated by aggregating the data at this level. This may skew the spatial pattern of the disease by averaging high and low sub-district areas. In the future, employing sub-district levels as a unit of analysis is advised, supposing higher spatial data resolution becomes available. 

## 6. Conclusions

The West Java social restrictions policy had a limited impact on the transmission of SARS-Cov-2, particularly in preventing the second pandemic wave in 2021, which was mainly driven by the circulation of the delta variant. A higher cumulative incidence was found in areas with a high population density and low vaccination rates, and these areas should be prioritized for geographically targeted mitigation efforts. 

## Figures and Tables

**Figure 1 ijerph-20-03198-f001:**
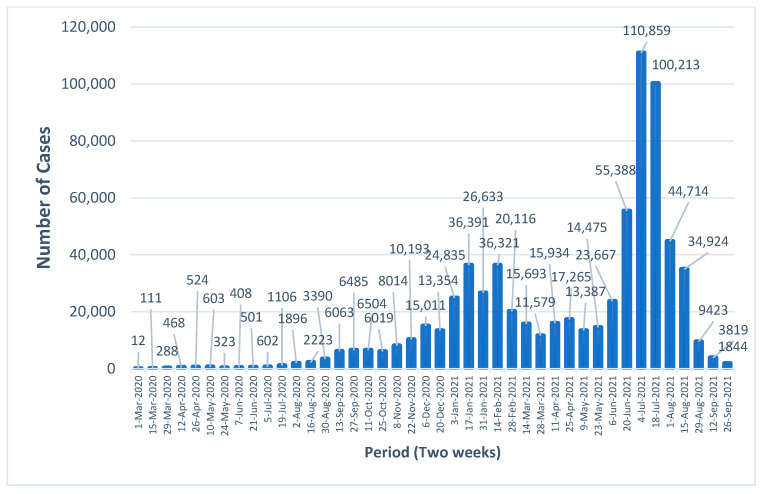
Bi-weekly number of new cases during March 2020–September 2021 in West Java.

**Figure 2 ijerph-20-03198-f002:**
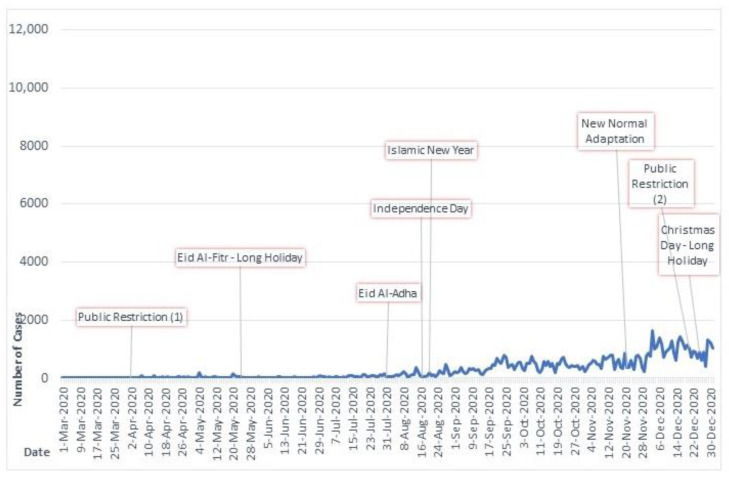
Daily number of new cases and events during March–December 2020 in West Java. Footnote: Public restrictions 1 and 2 are national policy, new normal adaptation is a relaxation of public restriction, office and outside home activities are allowed with limitations.

**Figure 3 ijerph-20-03198-f003:**
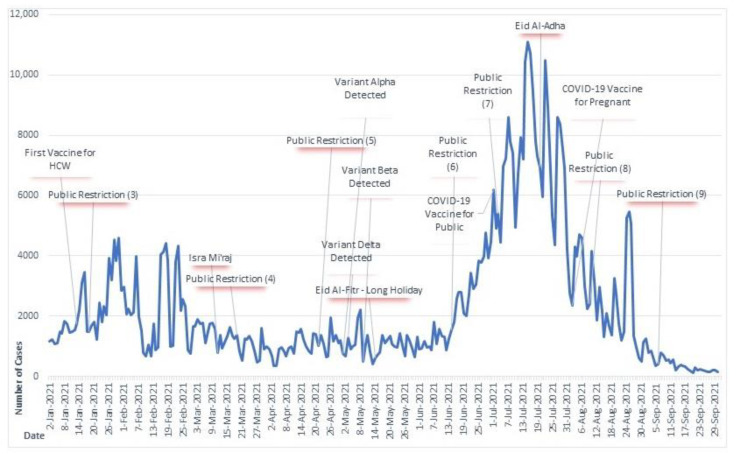
Daily number of new cases and events during January–September 2021 in West Java. Footnote: Public restrictions 3 and 7 are province policy. Public restrictions 4, 5, and 6 are national policies. Public restrictions 8 and 9 are national policies in some areas.

**Figure 4 ijerph-20-03198-f004:**
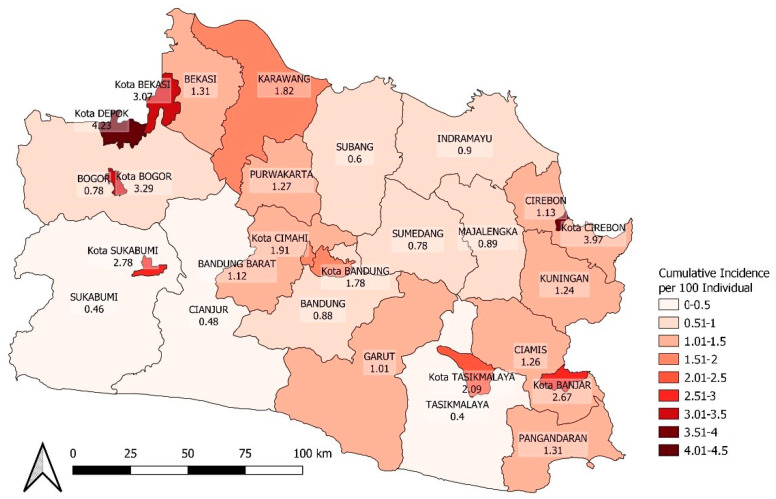
Spatial heterogeneity of COVID-19 incidence from March 2020 to September 2021 in West Java.

**Figure 5 ijerph-20-03198-f005:**
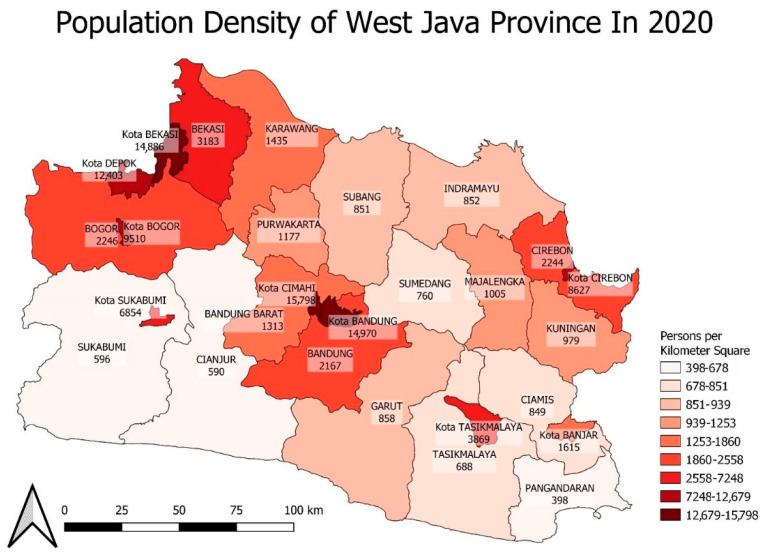
Spatial population density heterogeneity in West Java.

**Figure 6 ijerph-20-03198-f006:**
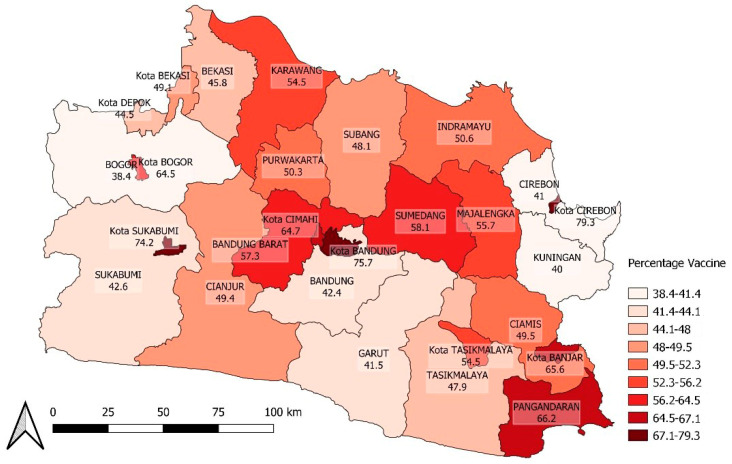
Spatial cumulative vaccination COVID-19 dose 1 coverage heterogeneity in West Java.

**Figure 7 ijerph-20-03198-f007:**
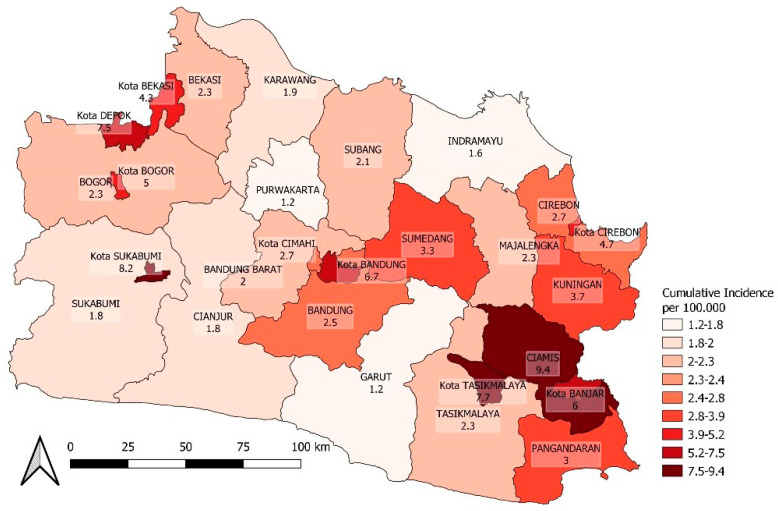
Spatial COVID-19 incidence heterogeneity for one week with a 30-day lag after vaccination.

**Figure 8 ijerph-20-03198-f008:**
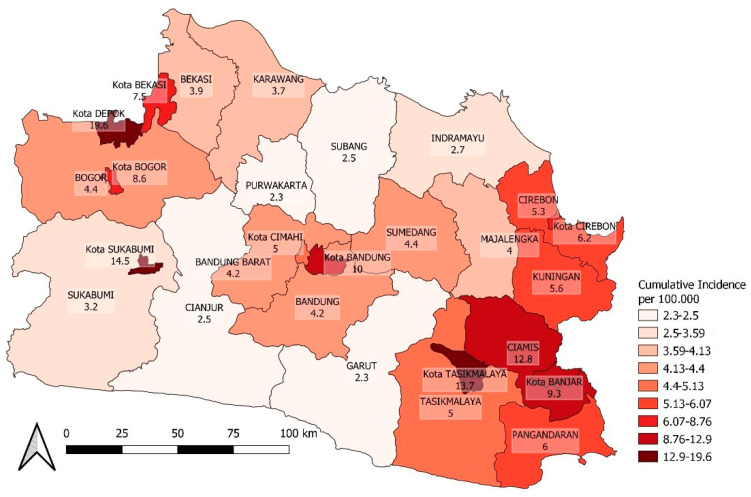
Spatial COVID-19 incidence heterogeneity for two weeks with a 30-day lag after vaccination.

**Figure 9 ijerph-20-03198-f009:**
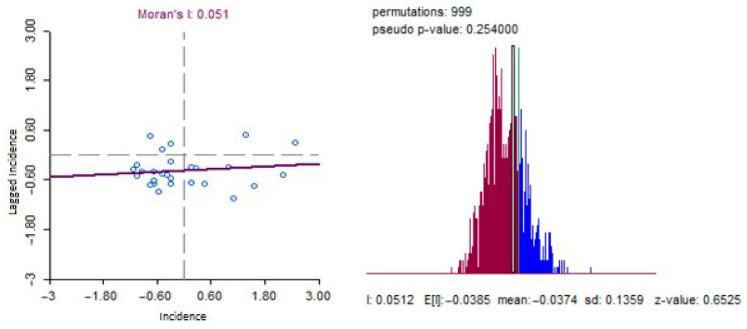
Scatterplot of global autocorrelation (Moran’s I) for COVID-19 incidence in West Java.

**Figure 10 ijerph-20-03198-f010:**
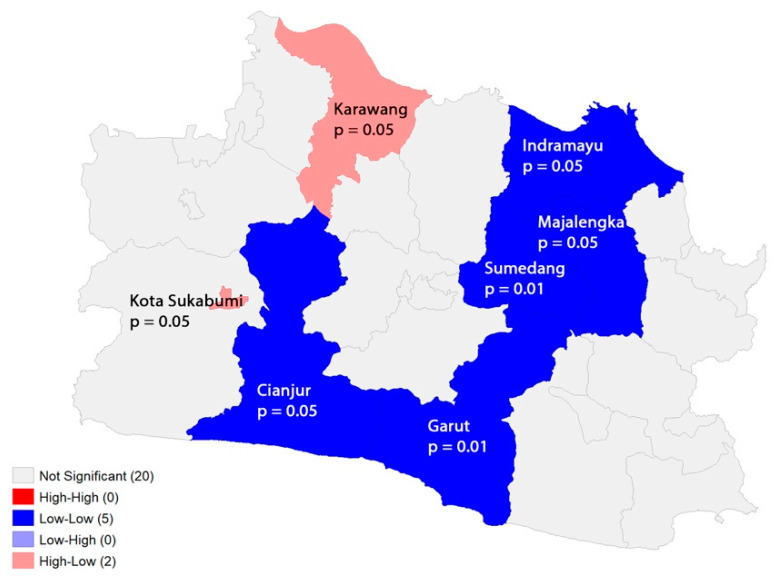
Local indicator of spatial association for COVID-19 cumulative incidence in West Java.

**Table 1 ijerph-20-03198-t001:** Descriptive statistics of variables in all districts.

	*n* = 27					
Variable	Min	Max	Median	IQR	Mean	SD
Population density	398	15,798	1435	4510	4100.85	4949.11
Cumulative vaccine coverage—dose 1 (%)	38.40	79.30	50.30	16.15	53.76	11.23
Total COVID-19 CI	0.40	4.30	1.30	1.10	1.62	1.06
One-week cumulative incidence (30 days lagged after vaccination)	1.21	9.40	2.67	2.77	3.70	2.31
Two-week cumulative incidence ((30 days lagged after vaccination)	2.26	19.56	4.96	4.25	6.41	4.27

**Table 2 ijerph-20-03198-t002:** Descriptive statistics of variables in low–low core clusters with a *p*-value = 0.01.

	District Area	
Variable	Garut	Sumedang
Population density	858 (1st Quartile)	760 (1st Quartile)
Cumulative vaccine coverage—dose 1 (%)	41.5 (2nd Quartile)	58.1 (2nd Quartile)
Total COVID-19CI	1	0.8
One-week cumulative incidence (30 days lagged after vaccination)	1.2	2.3
Two-week cumulative incidence (30 days lagged after vaccination)	3.3	4.4

**Table 3 ijerph-20-03198-t003:** Multivariable linear regression analysis of COVID-19 incidence in West Java.

Variable	Model 1 Outcome Variable: Total Cumulative Incidence		Model 2 Outcome Variable: One-Week Cumulative Incidence (30 days Lagged after Vaccination)		Model 3 Outcome Variable: Two-Week Cumulative Incidence (30 days Lagged after Vaccination)	
	Beta	*p*-Value	Beta	*p*-Value	Beta	*p*-Value
Population density	**0.0001**	**<0.01**	**0.008**	**<0.01**	**0.02**	**<0.01**
Cumulative vaccine coverage—dose 1 (%)	0.02	0.13	**−2.5**	**<0.01**	**−5.6**	**<0.01**
R-squared	0.53		0.45		0.48	
Model fit (AIC)	65		275		313	

## Data Availability

Not applicable.

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
