# Peer review of "Spatial and Temporal Analysis of COVID-19 Cases in West Java, Indonesia and Its Influencing Factors"

_ijerph, 2023, doi:10.3390/ijerph20043198_

Round 1

Reviewer 1 Report

Summary: Authors used spatial and temporal analysis of COVID-19 in West Java to understand regional distribution pattern of the disease and to plan for strategies to control the pandemic.

Major comments:

1. Authors did not mention how their analysis are different from previous studies using spatial and temporal analysis.

2. The authors did not consider the impact of different variants of virus in their analysis.

3. The spatial and temporal methods have not been explained adequately. 

Author Response

English editing and many times reading has been done
Hopefully it can clarify the delivery and provide comfort in reading.

  1. We added up  reference in our country, Previous studies did not use a spatial analysis to search for a global and local cluster, just disease mapping to see the distribution of cases. We use vaccine and population density to examine the impact of factors that cause dynamic cases in COVID-19. In a previous study, temporal analysis did not include an event or holiday to see people's interactions, which make a dynamic cases. In the study, we use cumulative incidents rather than individual cases to gain a better understanding of the burden.
  2. We have included the detection time of the new virus variant in a temporal chart and explained where it would make a dynamic case.

  3. We have already provided a more comprehensive explanation of spatial and temporal study in this updated document.
    We Already make it more clear and easy to understand.

Reviewer 2 Report

1-     The aim of the research should be clearly mentioned in the last paragraph of the introduction.

2-     The introduction needs some improvements. I suggest adding the following articles to the introduction. The first two ones are systematic reviews strengthening your introduction and the last one is an original related article to your research.

https://www.mdpi.com/1660-4601/18/22/12024

3-     I don’t know the relevance of section 2.1 (Immunity of COVID-19) to this article.

4-     Section 2.2 is not a methodology part, I think it can be removed form methodology part and after summarization, it should be mixed with the current introduction of the manuscript.

5-     The article’s English should be edited. For example this part is not correct English:

“Then, perform a spatial auto- 112 correlation analysis to examine the relationship between the regions based on Moran's I 113 index and local autocorrelation that measures cluster data locally.”

6-     (Lines 117-118) Geoda can implement different Regression analysis. The authors should describe the regression used clearly.

7-     I think Figure 1 has a problem. The numbers in the legend are strange. They cannot be cumulative incidence per 100 people. Are they the absolute number of patients?

8-     I would like to see the author’s revision and then review the rest of the paper. 

Author Response

We do your suggestion revision and improve paper such as adding reference, picture/figure, English editing and many times reading has been done
Hopefully it can clarify the delivery and provide comfort in reading.

thank you in advance
